# Multi-task Learning for Brain Network Analysis in the ABCD Study

Xuan Kan
*Department of Computer Science*
*Emory University*
Atlanta, US
xuan.kan@emory.edu

Hejie Cui
*Department of Computer Science*
*Emory University*
Atlanta, US
hejie.cui@emory.edu

Keqi Han
*Department of Computer Science*
*Emory University*
Atlanta, US
keqi.han@emory.edu

Ying Guo
*Department of Biostatistics and Bioinformatics*
*Emory University*
Atlanta, US
yguo2@emory.edu

Carl Yang
*Department of Computer Science*
*Emory University*
Atlanta, US
j.carlyang@emory.edu

*Abstract*—The Adolescent Brain Cognitive Development study provides a rich data resource for exploring the associations between brain network and cognitive, personality, and mental health measures in adolescents. To leverage this rich dataset, we propose a novel multi-task learning framework that predicts these measures from multi-view brain network data using a graph transformer architecture. Our approach learns shared representations across tasks while allowing for task-specific predictions, improving performance compared to single-task learning. Ablation studies reveal the importance of our proposed techniques of Batch-Wise Loss Balancing and Target Standardization in ensuring successful multi-task learning. Furthermore, we develop innovative visualization techniques based on integrated gradients to interpret the learned task correlations and identify influential brain network edges for each task. Our findings contribute to understanding the complex relationships between brain connectome and behavioral outcomes, highlighting the potential of multi-task learning in this domain. The implementation is available at https://github.com/Wayfear/MTML-ABCD/.

This research was partially supported by the National Science Foundation under Award Numbers 2319449 and 2312502, as well as the National Institute of Health under Award Numbers R01MH105561 and R01MH118771.

Data used in the preparation of this article were obtained from the Adolescent Brain Cognitive Development (ABCD) Study (https://abcdstudy.org), held in the NIMH Data Archive (NDA). This is a multisite, longitudinal study designed to recruit more than 10,000 children age 9-10 and follow them over 10 years into early adulthood. The ABCD Study® is supported by the National Institutes of Health and additional federal partners under award numbers U01DA041048, U01DA050989, U01DA051016, U01DA041022, U01DA051018, U01DA051037, U01DA050987, U01DA041174, U01DA041106, U01DA041117, U01DA041028, U01DA041134, U01DA050988, U01DA051039, U01DA041156, U01DA041025, U01DA041120, U01DA051038, U01DA041148, U01DA041093, U01DA041089, U24DA041123, U24DA041147. A full list of supporters is available at https://abcdstudy.org/federal-partners.html. A listing of participating sites and a complete listing of the study investigators can be found at https://abcdstudy.org/consortium_members/. ABCD consortium investigators designed and implemented the study and/or provided data but did not necessarily participate in the analysis or writing of this report. This manuscript reflects the views of the authors and may not reflect the opinions or views of the NIH or ABCD consortium investigators. The ABCD data repository grows and changes over time. The ABCD data used in this report came from NIMH Data Archive Release 4.0 (DOI 10.15154/1523041). DOIs can be found at https://nda.nih.gov/abcd.

*Index Terms*—Brain Networks, Deep Learning, Multi-task Learning, fMRI

## I. INTRODUCTION

Adolescent Brain Cognitive Development (ABCD) study [1] is the largest and most long-term study of brain development and child health in the US. It provides a vast brain development dataset in a diverse population, including functional magnetic resonance imaging data (fMRI) and abundant biological and behavioral survey results. This dataset offers an opportunity to explore the relationship between intricate brain connections and various behavioral data [2]–[7].

Leveraging the potential of neuroimaging data, recent studies have shown a growing trend of using brain networks derived from fMRI to predict various clinical outcomes and individual behaviors with different models [8]–[16]. Researchers have also developed innovative approaches to analyze these models and uncover potential correlations between functional brain networks and predicted outcomes [17]–[23]. For example, Kawahara et al [9] introduced BrainNetCNN, a convolutional neural network designed to predict cognitive and motor developmental outcome scores from brain networks. Similarly, Li et al [10] proposed a graph neural network model to predict clinical targets and discovered task-specific neurological biomarkers, demonstrating the effectiveness of graph-based approaches in capturing meaningful patterns in brain networks. Chen et al [24] further extended this line of research by training kernel regression models for 36 tasks and analyzing task relationships based on the learned model. These studies highlight the potential of leveraging brain networks to gain insights into the underlying neural mechanisms associated with various clinical outcomes and individual behaviors.

Multi-task learning (MTL) [25]–[28] has emerged as a promising approach for improving the generalization abilities of predictive models by enabling multiple learning tasks to share their knowledge. In the context of brain network analysis

in ABCD, where there is a diverse range of prediction targets, MTL can be particularly beneficial. By training several tasks simultaneously, MTL allows for a more native capture of task correlations, potentially leading to improved individual task performance. This contrasts with the approach taken by Chen et al [24], which builds individual models for each behavior task. By leveraging the shared representations learned across tasks, MTL can enhance the model's ability to uncover the underlying relationships between brain networks and various behavioral measures, resulting in more accurate and generalizable predictions.

Our study employs a shared bottom multi-task learning (MTL) approach for brain network analysis using the ABCD dataset, predicting multiple behavioral outcomes from resting-state functional connectivity. This differs from Xiao et al.'s work [29], which uses a manifold regularized MTL model to predict IQ scores from two task-based fMRI paradigms. While both leverage MTL for fMRI analysis, our study addresses a broader range of prediction tasks and uses resting-state data, potentially capturing more general brain organization patterns. In contrast to Marquand et al.'s Bayesian MTL framework for multi-subject decoding [30], our approach focuses on learning shared representations across tasks rather than subjects. We use a neural network instead of a Bayesian framework, which may offer computational efficiency advantages but potentially at the cost of explicit uncertainty quantification. Like ours, Huang et al.'s study uses resting-state fMRI data for MTL, but employs a Multi-cluster Multi-gate Mixture-of-Experts (M-MMOE) model for joint diagnosis of ASD and ADHD [31]. Our shared bottom model provides a simpler, more interpretable approach, while the M-MMOE allows for more complex task relationships. Additionally, our study encompasses a broader range of behavioral measures beyond just ASD and ADHD diagnosis, potentially offering insights into a wider spectrum of brain-behavior relationships.

In this work, we propose a novel MTL framework that jointly trains 35 tasks using multi-view functional brain networks from 6,682 samples in the ABCD study. We employ the Brain Network Transformer [32] as the backbone model, which converts a brain network into a graph-level embedding. This embedding is then fed into task-specific fully connected networks for each prediction target. By learning shared representations across tasks while still allowing for task-specific predictions, our approach aims to leverage the commonalities between tasks and improve overall prediction performance. Our main contributions are summarized as follows:

- We propose a novel MTL framework for predicting various measures from multi-view brain network data using a graph transformer architecture. Our approach learns shared representations across tasks while allowing for task-specific predictions, improving performance compared to single-task learning. Besides, the ablation study shows the effectiveness of two key training strategies during Multi-Task training.
- We conduct extensive experiments on the ABCD dataset, including 35 tasks categorized into three domains: *cogni-

tion*, *personality*, and *mental health*. We demonstrate the impact of MTL on different types of tasks.
- We develop innovative visualization techniques based on integrated gradients to interpret the learned task correlations and identify influential brain network edges, contributing to a better understanding of the complex relationships between brain connectome and behavioral outcomes.

## II. METHOD

### A. Problem definition

Let $\mathcal{D} = \{(\mathbf{X}^{(i)}, \mathbf{Y}^{(i)})\}_{i=1}^n$ be a dataset consisting of $n$ samples. For each sample $i$, the input $\mathbf{X}^{(i)} = \{X_1^{(i)}, X_2^{(i)}, \ldots, X_v^{(i)}\}$ represents a collection of $v$ brain networks, each derived from a distinct fMRI task (e.g., resting-state, stop-signal task, and N-Back). These brain networks, denoted by $X_j^{(i)} \in \mathbb{R}^{M \times M}$, capture the functional connectivity between $M$ brain regions. The corresponding prediction target $\mathbf{Y}^{(i)} = \{y_1^{(i)}, y_2^{(i)}, \ldots, y_t^{(i)}\}$ is a set of $t$ behavioral measures, such as cognitive scores, personality traits, or mental health indicators, associated with the $i$-th subject. In short, given this multi-view, multi-task dataset, our goal is to develop a predictive model that leverages the complementary information from the $v$ brain networks to simultaneously predict the $t$ behavioral outcomes.

### B. Model Architecture

Figure 1 shows the proposed multi-task learning framework for predicting behavioral outcomes from brain networks.

**Shared Representation Learning.** The footstone of our framework is Brain Network Transformer (BNT) [32], which serves as the shared backbone model. BNT is designed to process individual views of brain networks, denoted as $X_j^{(i)}$, where $j \in \{1, \ldots, v\}$ indexes the view and $i \in \{1, \ldots, n\}$ indexes the sample. For each view $j$, BNT learns a hidden representation embedding $h_j^{(i)} = \text{BNT}(X_j^{(i)})$. These view-specific embeddings capture the patterns present in the corresponding brain networks derived from distinct fMRI tasks. To obtain a comprehensive representation of each sample, we concatenate the view-specific embeddings $h_j^{(i)}$ from all $v$ views, resulting in a sample-level embedding $H^{(i)} = \bigoplus_{j=1}^v h_j^{(i)}$, where $\bigoplus$ denotes the concatenation operation. This sample-level embedding integrates the multifaceted information captured across different fMRI views, providing a holistic representation of each individual's brain connectivity patterns.

**Task-specific Prediction.** To achieve multi-task learning, we employ a separate Multi-Layer Perceptron (MLP) for each task $k \in \{1, \ldots, t\}$. These task-specific MLPs take the sample-level embedding $H^{(i)}$ as input and predict the corresponding behavioral outcome $\hat{y}_k^{(i)} = \text{MLP}_k(H^{(i)})$. By leveraging dedicated MLPs for each task, our framework allows for task-specific adaptations while benefiting from the shared representation learned by the BNT.

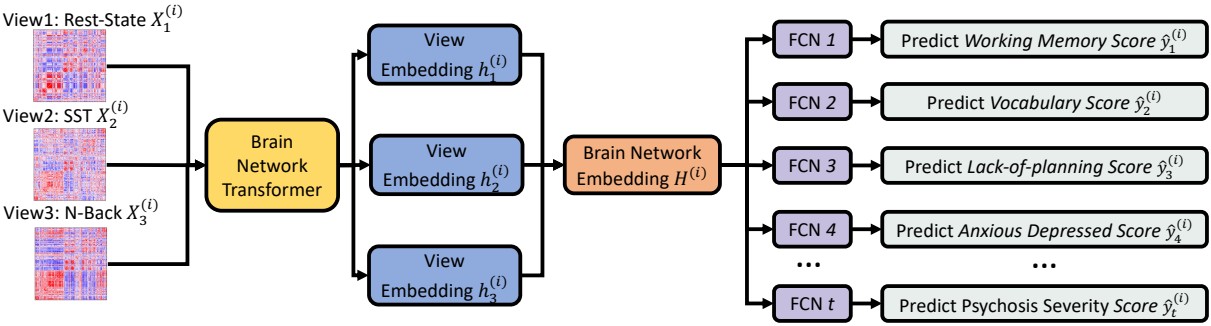

Fig. 1. Overview of our multi-task learning framework for predicting various measures from multi-view brain networks. Given a set of brain networks $\{X_1^{(i)}, X_2^{(i)}, \ldots, X_v^{(i)}\}$ derived from different views for the $i$-th subject, the Brain Network Transformer generates a unified brain network embedding $H^{(i)}$. This embedding is then fed into task-specific FCN to predict the corresponding target scores $\{\hat{y}_1^{(i)}, \hat{y}_2^{(i)}, \ldots, \hat{y}_t^{(i)}\}$ for various measures. The entire framework is trained end-to-end using multi-task learning, allowing for the sharing of knowledge across tasks while still enabling task-specific predictions.

## C. Multi-task Training Strategies

The entire framework is trained end-to-end using a multi-task learning approach. However, training a model to simultaneously predict multiple behavioral outcomes presents challenges due to the diverse characteristics and varying scales of the prediction targets. We introduce two key strategies to address these challenges and ensure effective training: Batch-Wise Loss Balancing and Target Standardization.

**Batch-Wise Loss Balancing.** In multi-task learning, tasks with larger loss values can potentially dominate the training process, hindering the model's ability to learn from all tasks equally. To mitigate this issue, we develop batch-wise loss balancing that adaptively adjusts the weight of each task's loss within a training batch. Let $L_k$ denote the loss associated with task $k$, where $k \in \{1, \ldots, t\}$. We compute the balanced loss $\hat{L}_k$ for each task as follows: $\hat{L}_k = \frac{L_k}{\bar{L}_k}$, where $\bar{L}_k$ is the average loss value for task $k$ over the current batch of samples. By normalizing each task's loss to have an average value of 1, we ensure that all tasks contribute equally to the overall optimization process. The total balanced loss $L_{\text{total}}$ is then calculated as the sum of all individual balanced task losses: $L_{\text{total}} = \sum_{k=1}^{t} \hat{L}_k$.

**Target Standardization.** Another challenge in multi-task learning is the varying scales of the target variables across different tasks. To address this issue, we employ a target standardization preprocessing step. For a regression task $k$, where $k \in \{1, \ldots, t\}$, we standardize the training labels $y_k^{(i)}$ to have zero mean and unit variance: $y_k^{(i)} = \frac{y_k^{(i)} - \mu_k}{\sigma_k}$, where $\mu_k$ and $\sigma_k$ denote the mean and standard deviation of the training labels for task $k$, respectively. This normalization brings all tasks to a similar scale, facilitating the model's ability to learn from them concurrently. During the validation and testing phase, we apply the inverse of this normalization process to transform the predicted labels $\hat{y}_k^{(i)}$ back to their original scale. Using the mean $\mu_k$ and standard deviation $\sigma_k$ computed from the training set, we perform the following operation: $\hat{y}_k^{(i)} = \hat{y}_k^{(i)} \cdot \sigma_k + \mu_k$. By standardizing the targets during training and reversing the normalization during inference, we ensure that the model can effectively learn from tasks with different scales while producing predictions in the original target domain.

## III. EXPERIMENTS

The following section describes our experimental methodology and findings, evaluating our multi-task learning framework for brain network analysis using the ABCD dataset. We detail our data, model architecture, training procedures, and evaluation metrics. We then present results comparing our multi-task approach to single-task baselines across various behavioral measures, followed by an ablation study on our key training strategies. Finally, we analyze task correlations learned by our model, providing insights into brain-behavior relationships in adolescent development.

### A. Dataset

We use the Adolescent Brain Cognitive Development (ABCD) dataset [1], which includes fMRI and behavioral data from a large cohort of children. We utilize resting-state and task-based fMRIs (stop-signal task [33] and N-Back task [34]) for brain network construction based on the HCP 360 ROI atlas [35]. In this study, we aim to predict 35 distinct labels, which span across 15 *neurocognitive ability* [36], 9 *impulsivity-related personality* and 11 *mental health assessments* [37], as detailed in Table I. The experimental dataset includes 6,682 samples with quality control procedures and filtering samples with incomplete fMRI and behavioral data.

### B. Setting

We employ a Brain Network Transformer as the shared model backbone, which consists of 3 transformer layers with 4 attention heads and an output dimension matching the number of nodes (360) in the brain network. The transformer layers are followed by task-specific 3-layer MLP branches with activation functions, each responsible for making predictions for one of the tasks. Since all tasks are regression tasks, we use Mean Squared Error (MSE) Loss as the loss function for all tasks. The model, which can predict 35 tasks simultaneously, has a total of 22.52 million parameters. We randomly split

TABLE I
SUMMARY OF TASKS AND THEIR DISTRIBUTIONS. FOR NUMERICAL
MEASURES, THE DISTRIBUTION IS PRESENTED AS MEAN±STANDARD
DEVIATION.

| Type | Task | ABCD field | Distribution |
|---|---|---|---|
| Cognition | Vocabulary | nihtbx_picvocab_uncorrected | 85.29±7.73 |
| | Attention | nihtbx_flanker_uncorrected | 94.56±8.72 |
| | Working Memory | nihtbx_list_uncorrected | 97.69±11.47 |
| | Executive Function | nihtbx_cardsort_uncorrected | 93.37±8.86 |
| | Processing Speed | nihtbx_pattern_uncorrected | 88.75±14.36 |
| | Episodic Memory | nihtbx_picture_uncorrected | 103.58±12.03 |
| | Reading | nihtbx_reading_uncorrected | 91.40±6.52 |
| | Fluid Cognition | nihtbx_fluidcomp_uncorrected | 92.61±10.20 |
| | Crystallized Cognition | nihtbx_cryst_uncorrected | 87.10±6.65 |
| | Overall Cognition | nihtbx_totalcomp_uncorrected | 87.28±8.56 |
| | Short Delay Recall | pea_ravlt_sd_trial_vi_tc | 9.88±2.95 |
| | Long Delay Recall | pea_ravlt_ld_trial_vii_tc | 9.40±3.10 |
| | Fluid Intelligence | pea_wiscv_trs | 18.19±3.71 |
| | Visuospatial Accuracy | lmt_scr_perc_correct | 0.60±0.17 |
| | Visuospatial Reaction time | lmt_scr_rt_correct | 2691.27±461.01 |
| Personality | Negative Urgency | upps_y_ss_negative_urgency | 8.40±2.61 |
| | Lack of Planning | upps_y_ss_lack_of_planning | 7.68±2.29 |
| | Sensation Seeking | upps_y_ss_sensation_seeking | 9.84±2.67 |
| | Positive Urgency | upps_y_ss_positive_urgency | 7.86±2.89 |
| | Lack of Perseverance | upps_y_ss_lack_of_perseverance | 6.96±2.17 |
| | Behavioral Inhibition | bis_y_ss_bis_sum | 9.45±3.66 |
| | Reward Responsiveness | bis_y_ss_bas_rr | 10.94±2.90 |
| | Drive | bis_y_ss_bas_drive | 3.96±2.97 |
| | Fun Seeking | bis_y_ss_bas_fs | 5.65±2.59 |
| Mental Health | Total Psychosis Symptoms | pps_y_ss_number | 2.30±3.30 |
| | Psychosis Severity | pps_y_ss_severity_score | 5.33±9.44 |
| | Anxious Depressed | cbcl_scr_syn_anxdep_r | 2.45±3.01 |
| | Withdrawn Depressed | cbcl_scr_syn_withdep_r | 0.97±1.64 |
| | Somatic Complaints | cbcl_scr_syn_somatic_r | 1.46±1.92 |
| | Social Problems | cbcl_scr_syn_social_r | 1.46±2.13 |
| | Thought Problems | cbcl_scr_syn_thought_r | 1.53±2.08 |
| | Attention Problems | cbcl_scr_syn_attention_r | 2.71±3.30 |
| | Rule-breaking Behavior | cbcl_scr_syn_rulebreak_r | 1.07±1.70 |
| | Aggressive Behavior | cbcl_scr_syn_aggressive_r | 3.02±4.11 |
| | Mania | pgbi_p_ss_score | 1.16±2.56 |

the ABCD dataset into training (70%), validation (10%), and testing (20%) subsets. During training, we use the Adam optimizer with a weight decay of $10^{-4}$, a cosine learning rate scheduler (initial: $10^{-4}$, final: $10^{-5}$), and a batch size of 16. The model is trained for 100 epochs, and the model whose epoch shows the best total loss on the validation set is selected as the final model to report performance.

### C. Metrics

To evaluate our multi-task learning model's performance on the 35 regression tasks from the ABCD dataset, we use two metrics: Mean Squared Error (MSE) and R-squared ($R^2$). MSE measures the average squared difference between predicted and actual values, with lower values indicating better performance. $R^2$, on the other hand, is used to compare performance across tasks with varying label scales. $R^2$ values range from $-\infty$ to 1, where negative values indicate worse performance than using the target variable's mean, zero indicates equivalence to using the mean, and positive values suggest the model captures useful information from brain networks and the prediction beats the mean of the target. Thus, $R^2$ can be used to evaluate a task's predictability. All reported results are averaged over 5 runs with different random seeds.

### D. Performance Evaluation

The overall results of our multi-task learning model on the ABCD dataset are shown in Table II. From the table, we can obtain 3 key insights: (1) Single-task performance: The **Single-Task** column reveals that for all Personality and Mental Health tasks, these models' $R^2$ is below 0.03, indicating that there is limited predictive power when using brain networks to predict these labels. In contrast, for the Cognition tasks, except for the Visuospatial Reaction Time task, all other **14** tasks have an $R^2$ greater than or equal to 0.03. This suggests that the model can capture useful information from the brain networks and outperform predictions based solely on the mean of the target variable for these Cognition tasks; (2) Multi-task learning benefits: By comparing the Single-Task performance column with the **Multi-Task** performance column, we observe that multi-task training improves the performance of almost all tasks that already exhibit predictive power in the single-task setting. However, the Personality and Mental Health tasks that were unpredictable in the single-task setting remain unpredictable in the multi-task setting, indicating that these tasks cannot effectively leverage useful information from other tasks during joint training; (3) Impact of removing unpredictable tasks: To further investigate the influence of the unpredictable *Personality* and *Mental Health* tasks on the overall model performance, we conducted an additional experiment where we removed these tasks during multi-task training. The results of this experiment are shown in the **Multi-CogTask** column. Interestingly, we observe that by excluding these unpredictable tasks, the performance of the remaining tasks drops compared to the multi-task setting that includes all tasks. This finding suggests that labeling information from Personality and Mental Health tasks is still helpful for other tasks, even though these tasks themselves remain unpredictable.

### E. Ablation Study

We investigate the effectiveness of our two key training strategies in our multi-task learning model: *Batch-Wise Loss Balancing* and *Target Standardization*. We compare the performance of our full model with three ablated versions: (1) without Batch-Wise Loss Balancing, (2) without Target Standardization, and (3) without both strategies. The results in Fig. 2 show that removing Batch-Wise Loss Balancing leads to a slight decrease in performance across all 14 predictable tasks, while removing Target Standardization causes a significant drop. When both strategies are removed, the model fails to learn any meaningful information, resulting in negative R² values for all tasks. This study demonstrates the importance of these training strategies in enabling successful multi-task learning for brain network analysis.

## IV. TASK CORRELATION ANALYSIS

In this section, we visualize the task correlations learned by our multi-task learning model using the integrated gradients method [38] and compare them with the inherent correlations between task labels. Finally, we visualize these important edges for prediction in different tasks.

### A. Generating Task Correlation Data

Algorithm 1 illustrates the process to obtain the task-level correlation matrix $C$ and edge importance $\mathbf{G}^k$ for each task $k$.

PERFORMANCE COMPARISON OF SINGLE-TASK, MULTI-TASK, AND MULTI-TASK (COGNITION TASKS ONLY) MODELS ON THE ABCD DATASET. TASKS WITHIN EACH TYPE (I.G., COGNITION, PERSONALITY AND MENTAL HEALTH) ARE SORTED IN DESCENDING ORDER BASED ON THE $R^2$ VALUE UNDER THE SINGLE-TASK COLUMN. TASKS HIGHLIGHTED IN PURPLE HAVE AN $R^2$ VALUE GREATER THAN OR EQUAL TO 0.03, AS THIS THRESHOLD INDICATES THAT THE MODEL CAPTURES MEANINGFUL INFORMATION FROM THE BRAIN NETWORKS AND OUTPERFORMS PREDICTIONS BASED SOLELY ON THE MEAN OF THE TARGET VARIABLE, THUS SIGNIFYING TASKS WITH NOTABLE PREDICTIVE POWER. **BOLD** VALUES INDICATE THE BEST RESULT FOR THESE PREDICTABLE TASKS ACROSS THE THREE MODEL SETTINGS. THE ↑ INDICATES A HIGHER METRIC IS BETTER, ↓ INDICATES A LOWER ONE IS BETTER.

| Type | Task | Single-Task | | Multi-Task | | Multi-CogTask | |
|---|---|---|---|---|---|---|---|
| | | MSE ↓ | $R^2$ ↑ | MSE ↓ | $R^2$ ↑ | MSE ↓ | $R^2$ ↑ |
| Cognition | OverallCognition | 54.34±2.35 | **0.26±0.02** | **53.79±2.27** | 0.24±0.04 | 58.30±3.15 | 0.20±0.03 |
| | CrystallizedCognition | **32.54±2.11** | **0.25±0.03** | 33.01±0.99 | 0.24±0.02 | 35.13±2.62 | 0.19±0.05 |
| | Vocabulary | 47.39±2.89 | 0.20±0.03 | **47.15±2.87** | **0.21±0.04** | 49.04±2.94 | 0.16±0.05 |
| | Reading | 35.57±1.79 | 0.15±0.03 | **34.17±0.83** | **0.15±0.03** | 36.85±2.61 | 0.14±0.03 |
| | FluidCognition | 89.97±2.24 | **0.14±0.03** | **87.87±4.55** | 0.12±0.05 | 92.50±3.95 | 0.11±0.02 |
| | FluidIntelligence | 12.56±0.32 | 0.12±0.01 | 12.28±0.25 | 0.12±0.03 | **12.03±0.60** | **0.13±0.02** |
| | WorkingMemory | 121.76±5.44 | 0.07±0.03 | **116.12±2.65** | **0.09±0.03** | 120.12±3.06 | 0.09±0.03 |
| | ExecutiveFunction | 72.55±3.74 | 0.07±0.01 | **70.33±2.33** | **0.07±0.02** | 73.09±3.98 | 0.07±0.02 |
| | ShortDelayRecall | 8.43±0.18 | 0.06±0.03 | **7.87±0.32** | **0.08±0.02** | 8.16±0.32 | 0.06±0.02 |
| | LongDelayRecall | 9.07±0.33 | 0.05±0.02 | **8.62±0.24** | **0.07±0.03** | 9.14±0.22 | 0.04±0.02 |
| | VisuospatialAccuracy | 0.03±0.00 | 0.05±0.04 | **0.03±0.00** | **0.09±0.02** | 0.03±0.00 | 0.08±0.01 |
| | Attention | 72.84±4.41 | 0.04±0.01 | 71.99±3.30 | 0.04±0.04 | **70.20±2.74** | **0.05±0.02** |
| | EpisodicMemory | 136.82±4.87 | 0.04±0.02 | **136.50±5.84** | **0.04±0.02** | 138.15±3.91 | 0.04±0.03 |
| | ProcessingSpeed | **198.17±5.99** | **0.03±0.01** | 200.26±8.89 | 0.02±0.02 | 203.91±8.01 | 0.00±0.03 |
| | VisuospatialReactionTime | 208k±7k | 0.00±0.00 | 212k±9k | 0.00±0.00 | 210k±4k | 0.01±0.01 |
| Personality | RewardResponsiveness | 8.44±0.37 | 0.01±0.00 | 8.33±0.51 | -0.02±0.05 | - | - |
| | Drive | 8.61±0.20 | 0.01±0.01 | 8.66±0.47 | 0.01±0.04 | - | - |
| | PositiveUrgency | 8.02±0.22 | 0.01±0.01 | 8.17±0.27 | -0.00±0.06 | - | - |
| | LackOfPlanning | 5.09±0.16 | 0.00±0.01 | 5.32±0.22 | -0.01±0.01 | - | - |
| | LackPerseverance | 4.86±0.11 | 0.00±0.01 | 4.54±0.19 | -0.01±0.01 | - | - |
| | FunSeeking | 6.62±0.16 | 0.00±0.00 | 6.72±0.22 | -0.01±0.04 | - | - |
| | SensationSeeking | 7.14±0.15 | -0.00±0.00 | 7.10±0.19 | -0.01±0.02 | - | - |
| | BehavioralInhibition | 13.48±0.10 | -0.01±0.01 | 13.50±0.71 | -0.02±0.03 | - | - |
| | NegativeUrgency | 6.90±0.18 | -0.01±0.03 | 6.85±0.39 | -0.02±0.07 | - | - |
| Mental Health | TotalPsychosisSymptoms | 11.21±0.69 | 0.01±0.01 | 10.76±0.90 | -0.01±0.04 | - | - |
| | AttentionProblems | 10.54±0.55 | 0.00±0.02 | 10.80±0.82 | -0.00±0.04 | - | - |
| | AnxiousDepressed | 8.68±0.59 | -0.00±0.00 | 8.67±0.84 | -0.04±0.06 | - | - |
| | AggressiveBehavior | 16.93±1.66 | -0.00±0.01 | 16.45±1.46 | -0.02±0.06 | - | - |
| | WithdrawnDepressed | 2.85±0.13 | -0.00±0.01 | 2.57±0.18 | -0.03±0.05 | - | - |
| | SomaticComplaints | 3.80±0.19 | -0.00±0.01 | 3.67±0.33 | -0.02±0.04 | - | - |
| | ThoughtProblems | 4.53±0.19 | -0.00±0.00 | 4.31±0.39 | -0.03±0.05 | - | - |
| | SocialProblems | 4.66±0.24 | -0.01±0.01 | 4.36±0.47 | -0.01±0.07 | - | - |
| | PsychosisSeverity | 87.75±7.75 | -0.01±0.04 | 89.23±8.65 | -0.00±0.04 | - | - |
| | Mania | 6.50±0.51 | -0.01±0.01 | 6.49±0.93 | -0.01±0.05 | - | - |
| | RuleBreakingBehavior | 3.01±0.26 | -0.02±0.03 | 2.75±0.15 | 0.01±0.04 | - | - |

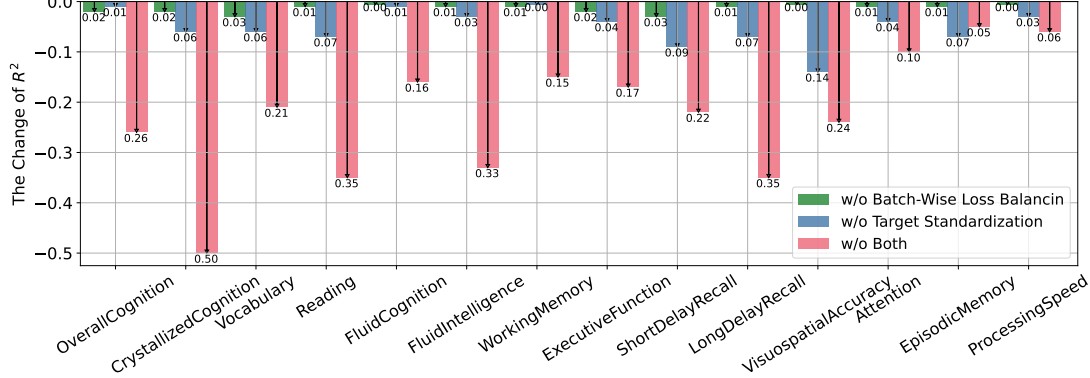

Fig. 2. Ablation study results comparing the performance of our full multi-task learning model with three ablated versions. The bars represent the difference in $R^2$ values between each ablated version and the full model for the 14 predictable tasks.

The algorithm uses these symbols: $\mathcal{M}$ for the trained multi-task learning model, $n_s$ for the number of samples selected from the test set, $\mathbf{X}^{(i)}$ and $\mathbf{Y}^{(i)}$ for the input brain networks and target labels of sample $i$, respectively, $v$ for the number

of views (e.g. rest-state or N-Back task), $M$ for the number of nodes in each brain network, $\mathbf{IG}_{j,e,k}^{(i)}$ for the integrated gradients of edge $e$ and task $k$ in view $j$ for sample $i$, $t$ for the total number of tasks, $\mathbf{I\bar{G}}_j^k$ for the average edge importance of view $j$ and task $k$, $\mathbf{g}_j^k$ for the vectorized edge importance of view $j$ and task $k$, $\mathbf{\hat{G}}^k$ for the concatenated edge importance across views for task $k$, $\mathbf{G}^k$ for the averaged edge importance across views for task $k$, and $\mathbf{C}$ for the correlation matrix between task-level edge importance vectors.

---

**Algorithm 1** Obtaining Task-level correlation matrix $C$ and edge importance $\mathbf{G}^k$ for each task $k$ by Integrated Gradients

---

**Require:** Trained multi-task learning model $\mathcal{M}$, test set $\mathcal{D}_{\text{test}}$, number of samples $n_s$
**Ensure:** Task-level correlation matrix $C$ and edge importance $\mathbf{G}^k$ for each task $k$
1: Save the best-performing model $\mathcal{M}^*$ based on validation set performance
2: Randomly select $n_s$ samples $\{(\mathbf{X}^{(i)}, \mathbf{Y}^{(i)})\}_{i=1}^{n_s}$ from $\mathcal{D}_{\text{test}}$
3: **for** each sample $(\mathbf{X}^{(i)}, \mathbf{Y}^{(i)})$ **do**
4:     **for** each view $j \in \{1, \ldots, v\}$ **do**
5:         **for** each edge $e \in \{1, \ldots, M^2\}$ **do**
6:             **for** each task $k \in \{1, \ldots, t\}$ **do**
7:                 Compute integrated gradients $\mathbf{IG}_{j,e,k}^{(i)}$ for edge $e$ in view $j$ and task $k$
8:             **end for**
9:         **end for**
10:     **end for**
11: **end for**
12: **for** each task $k \in \{1, \ldots, t\}$ **do**
13:     **for** each view $j \in \{1, \ldots, v\}$ **do**
14:         $\mathbf{I\bar{G}}_j^k = \frac{1}{n_s} \sum_{i=1}^{n_s} \mathbf{IG}_{j,k}^{(i)}$     $\triangleright$ Average edge importance across samples
15:     **end for**
16:     $\mathbf{\hat{IG}}^k = \bigoplus_{j=1}^v \mathbf{g}_j^k$    $\triangleright$ Concatenate edge importance across views for task $k$
17:     $\mathbf{G}^k = \frac{1}{v} \sum_{j=1}^v \mathbf{I\bar{G}}_j^k$    $\triangleright$ Average IG across views for task $k$
18: **end for**
19: Compute correlation matrix $\mathbf{C} \in \mathbb{R}^{t \times t}$ between $\{\mathbf{\hat{IG}}^k\}_{k=1}^t$

---

### B. Task Correlation Results

The correlation matrix $C$ (Fig. 3 Left) reveals the similarities between tasks based on the brain network edges that the model deems important for making predictions. In other words, tasks that have a high positive correlation in this matrix rely on similar brain network edges for their predictions, suggesting that the model has learned shared patterns for these tasks. This matrix provides insights into the model's internal representation and how it leverages the brain network information for different tasks.

On the other hand, the label correlation matrix (Fig. 3 Right) represents the inherent relationships between the task variables themselves, independent of the model's learning process. This matrix shows the correlations between the target variables of different tasks, revealing the intrinsic similarities or differences among them. By comparing these two matrices, we can assess the model's ability to capture meaningful task relationships from the brain network data. The results highlight two key findings: (a) Strong positive correlations among the 14 most predictable tasks: The model learns shared patterns for these tasks, as evidenced by the high positive correlations in the integrated gradients-based task correlation matrix. This suggests that the model has successfully identified common brain network edges that are predictive of these tasks, aligning with their inherent similarities revealed in the label correlation matrix; (b)Distinct patterns for personality and mental health task groups: Within each of these groups, tasks exhibit strong positive correlations in the integrated gradients-based task correlation matrix, indicating that the model learns shared patterns for tasks within the same group. However, the correlations between these two groups are weak, suggesting that the model learns distinct patterns for personality and mental health tasks. This finding implies that the model has captured the unique brain network edges that are relevant for each group of tasks, reflecting their underlying differences.

These observations demonstrate that our multi-task learning model effectively captures both the shared patterns among similar tasks and the distinct patterns between different task groups. By learning these task relationships from the brain network data, the model can leverage the commonalities among tasks to improve its predictions while still maintaining the ability to capture task-specific patterns. This highlights the power of our multi-task learning approach in uncovering meaningful relationships and utilizing them for predictions.

Furthermore, the alignment between the integrated gradients-based task correlation matrix and the label correlation matrix validates the model's interpretability. The model's learned task relationships, as revealed by the important brain network edges, match the inherent similarities and differences among the task variables. This interpretability strengthens the trustworthiness of our model and its potential for application in clinical and research settings.

### C. Visualization of Important Edges

To further investigate the task-specific patterns learned by our multi-task learning model, we visualize the most important edges for a subset of tasks in Figure 4. These edges are determined based on the integrated gradients $G^k$ obtained for each task $k$. Specifically, we select the top 0.05% of edges with the highest integrated gradient magnitudes and visualize them on a brain network template using BrainNet Viewer [39]. The node colors in the visualizations represent different functional modules, while the edge colors and thickness indicate the magnitude of the integrated gradients.

Figure 4 showcases the important edges for six tasks: two from the Cognition group (Vocabulary and Reading), two from the Personality group (Drive and Fun Seeking), and two from the Mental Health group (Rule-breaking Behavior and Aggressive Behavior). By comparing the visualizations

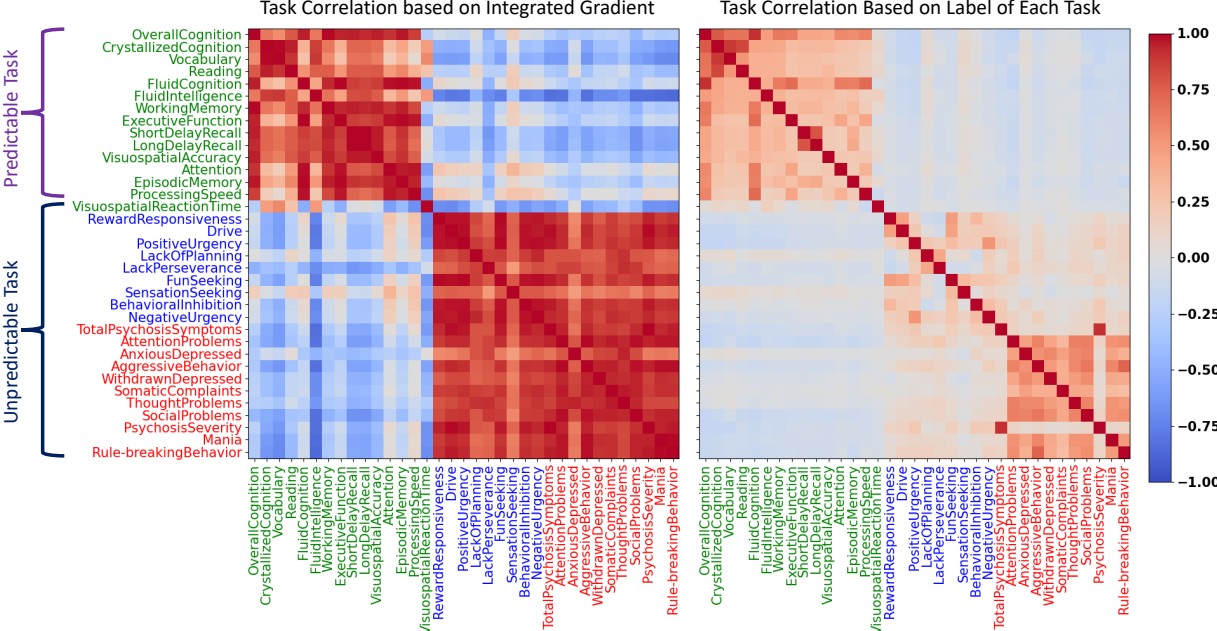

Fig. 3. Task correlation matrices based on integrated gradients (left) and task labels (right). The integrated gradients matrix reveals the correlation among tasks regarding their importance to the model's predictions, while the label correlation matrix shows the inherent relationships among task labels. The task names are color-coded based on their type: green for cognition, blue for personality, and red for mental health. The comparison of these two matrices provides insights into the model's ability to capture meaningful task relationships from data.

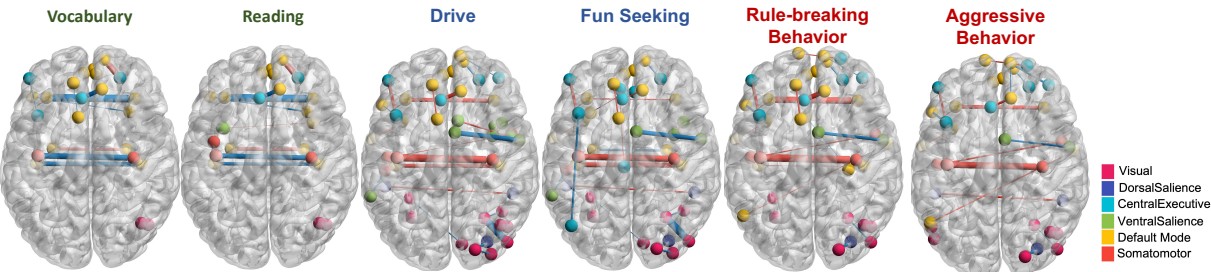

Fig. 4. Visualization of the top 0.05% brain network edges for 6 tasks, determined by integrated gradients $\mathbf{G}^k$. Node color indicates functional module, while edge color (blue for negative, red for positive) and thickness represent integrated gradient magnitude. This figure reveals key brain edges the model relies on for predictions in each task.

within and across task groups, we observe several interesting patterns. First, tasks within the same group tend to have similar important edges. For example, the Vocabulary and Reading tasks, both belonging to the Cognition group, share many common edges in the visualized brain networks. Similarly, the Drive and Fun Seeking tasks from the Personality group also exhibit overlapping important edges. This observation aligns with the within-group correlations depicted in the integrated gradients-based task correlation matrix (Figure 3, Left).

On the other hand, when comparing tasks from different groups, we notice distinct patterns in the important edges. For instance, the important edges for the Vocabulary task (Cognition group) differ significantly from those of the Drive task (Personality group). This finding suggests that our multi-task learning model captures task-specific patterns while also learning shared representations within task groups.

The visualization of important edges provides valuable insights into the brain network regions and connections that the model relies on for making predictions in each task.

## V. CONCLUSION

We proposed a novel multiple-task learning framework for predicting cognitive, personality, and mental health measures from brain networks using the ABCD dataset. Our approach effectively captures meaningful relationships across tasks and improves prediction performance compared to single-task learning. Through experiments, we demonstrated the importance of two training strategies and provided deep task correlation analysis by the integrated gradient method, presenting a significant step towards understanding the complex relationships between brain connectome and behavioral outcomes.

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
