# OpenReview forum: "Multi-task Learning for Brain Network Analysis in the ABCD study"
_IEEE.org/EMBS/BHI/2024/Conference — IEEE BHI'24_

### Official Review · Reviewer_Z8fY · 2024-07-25
**Multi-task Learning for Brain Network Analysis in the ABCD study**

**Overall Rating:** 8
**Confidence:** 4

**Other Quality Metrics:**

(a) Clarity of writing: great
(b) Clinical Significance: good
(c) Methodological Novelty: good
(d) Experiments and Results: great

**Questions For The Authors:**

- Why Chen et al. target 36 tasks while 35 tasks are investigated in this work?
- How was performed the Single-Task strategy?

**Strengths:**

Main contributions of this work are:
- demonstration of MTL effects for outcomes predictions
- integration and evaluation of two strategies to counteract the effects of different scales and the taking of tasks from others due to higher cost valuesfor during training
- a new visualisation technique for better interpretation of model behavior

The paper is well written, undertook experiments and results are correct and cleary explained.

**Summary Of The Paper:**

In this paper, authors propose to study multi-task learning (using a graph-based architecture) to predict cognition, personality and mental health outcomes from functionnal magnetic resonance imaging data of the Adolescent Brain Cognitive Development study.

**Weaknesses:**

The paper may lack of positioning according to the current state of art on multitask learning from fMRI and comparison/discussion with other techniques.

There is some redundancies in the model architecture part regarding the introduction of the Brain Network Transformer (BNT). I would suggest to merge "Brain Network Transformer" and "Shared Representation Learning" paragraph

It may be more relevant to put author's methodological contributions (resumed in the strenghts part) in evidence into the method section.

Minor comments :
- Add a sentence to explain how scores were defined in the ABCD study.
- Remove "our ablation study results in Section III-E show the necessity of incorporating them during training"
- Add an introductive paragraph to explain the content of the experiments section
- Precise on which approach was based the R^2 highlighting in Table II

---

### Official Review · Reviewer_1dTN · 2024-07-28
**Multi-task Learning for Brain Network Analysis in the ABCD study**

**Overall Rating:** 7
**Confidence:** 4

**Other Quality Metrics:**

(a) Clarity of writing: excellent
(b) Clinical Significance: great;
(c) Methodological Novelty: good;
(d) Experiments and Results: great

**Questions For The Authors:**

none

**Strengths:**

The paper is well-written and easy to follow. The visualization of the most important edges for different tasks is particularly helpful for model interpretation.

**Summary Of The Paper:**

This work presents a multi-task learning framework for predicting 35 tasks using fMRIs acquired from the Adolescent Brain Cognitive Development (ABCD) study. Transformers were employed as the backbone for shared feature learning, while multiple layer perceptrons were used for individual task prediction. Additionally, a visualization technique was developed for the interpretation of learned task correlations.

**Weaknesses:**

While the application to the ABCD study is new, the multi-task learning approach presented in this work appears to be quite standard, limiting the overall innovation.

---

> ### Author Rebuttal · Authors · 2024-09-02
>
> Dear Reviewer,
>
> Thank you for your thoughtful feedback on our paper. We appreciate your observation regarding the innovation of our multi-task learning approach. We'd like to address this point and provide some additional context.
> While we acknowledge that the core multi-task learning framework we employed is based on established methods, we believe our work offers several novel contributions:
>
> 1. Novel application: As you noted, this is the first application of multi-task learning to a large scale brain network datasets for predicting cognitive, personality, and mental health measures from brain networks. This novel use case provides valuable insights into adolescent brain development and mental health.
> 2. Task-specific adaptations: We made several adaptations to tailor the multi-task learning approach to the unique challenges of brain network analysis and the specific tasks at hand. These include our custom loss balancing strategy and target standardization techniques.
> 3. Integrated gradient analysis: Our use of integrated gradients for deep task correlation analysis is a novel contribution that provides interpretability and insights into the relationships between tasks and brain regions.
>
> While the core multi-task learning framework may be considered standard, we believe the integration of these elements, along with the specific application and insights gained, make significant contributions  to the field.

---

### Official Review · Reviewer_weg9 · 2024-07-31
**It is developed and implemented a multi-task learning framework for brain network analysis using the ABCD dataset. This framework predicts cognitive, personality, and mental health measures by capturing meaningful inter-task relationships and outperforming single-task learning models.**

**Overall Rating:** 8
**Confidence:** 4

**Other Quality Metrics:**

(a) Clarity of writing – Great
(b) Clinical Significance - Great
(c) Methodological Novelty - Great
(d) Experiments and Results - Great

**Questions For The Authors:**

Noting to point out.

**Strengths:**

In the field of brain network analysis, multi-task learning models hold great promise due to their ability to simultaneously address multiple related tasks, leading to improved performance and more comprehensive insights. To harness the full potential of multi-task learning models, it is crucial to implement effective training strategies. In this study, the authors focus on two key strategies: batch-wise loss balancing and target standardization. Their findings underscore the significance of these methods in achieving successful multi-task learning for brain network analysis.

**Summary Of The Paper:**

In this work, it is proposed a novel multi-task learning framework utilizing the ABCD dataset to predict cognitive, personality, and mental health measures from brain networks. This approach captures meaningful inter-task relationships and outperforms single-task learning models.  It is demonstrated the importance of two key training strategies and employed integrated gradient methods for deep task correlation analysis.

**Weaknesses:**

The authors recognize that the correlations between the two groups are weak, indicating that the model learns distinct patterns for personality and mental health tasks. This suggests that the model has identified unique brain network connections relevant to each group of tasks, highlighting their underlying differences.

---

> ### Author Rebuttal · Authors · 2024-09-02
>
> Dear Reviewer,
>
> Thank you for your thorough and insightful review of our paper on multi-task learning for brain network analysis. We greatly appreciate the time and effort you dedicated to evaluating our work.
> We appreciate your perspective on how the results highlight the model's ability to identify distinct patterns for different task groups. This is an intended design though and we will further develop techniques to reduce the negative effects brought by such differences.

---

### Decision · Program_Chairs · 2024-09-23

Accept